# An Assessment of Starch Content and Gelatinization in Traditional and Non-Traditional Dog Food Formulations

**DOI:** 10.3390/ani12233357

**Published:** 2022-11-30

**Authors:** Erin Beth Perry, Alyssa Ann Valach, Jesse Marie Fenton, George E. Moore

**Affiliations:** 1Department of Animal Science, Food and Nutrition, Southern Illinois University, Carbondale, IL 62901, USA; 2Department of Veterinary Administration, Purdue University, West Lafayette, IN 47907, USA

**Keywords:** dog food, starch, novel ingredient, starch gelatinization

## Abstract

**Simple Summary:**

The use of novel ingredients is increasing in popularity within the pet food industry. Products utilizing both grain-free and more traditional ingredient profiles are widely available. Unfortunately, gastrointestinal impacts associated with changes in starch content and gelatinization from different grain or carbohydrate sources are unknown. The purpose of this work was to measure starch content and gelatinization in commercially manufactured dry dog food products and assess differences related to ingredient profile. Gelatinized starch was proportionately lower in products manufactured with traditional ingredients (chicken, chicken by-product meal, meat and bone meal and plant-based ingredients including rice, barley, oats, and corn). Diets manufactured with non-traditional ingredients including alligator, buffalo, venison, kangaroo, squid, quail, rabbit, and salmon along with plant-based ingredients including tapioca, chickpeas, lentils, potato, and pumpkin had higher proportions of starch gelatinized. The degree of gelatinization can impact digestibility in the canine digestive tract and is an important area requiring further investigation.

**Abstract:**

Starch gelatinization in pet food may be affected by moisture, retention time, and ingredients used. Starch gelatinization has been associated with changes in digestibility but is not well studied using non-traditional ingredients in canine diets. The objective of this research was to examine differences in starch content and gelatinization associated with changes in ingredient profile (traditional vs. non-traditional) and nutrient content requirements associated with differing life stages. Traditional diets (*n* = 10) utilizing protein sources including chicken, chicken by-product meal, meat and bone meal and plant-based ingredients including rice, barley, oats, and corn were examined in comparison with non-traditional diets (*n* = 10) utilizing protein sources including alligator, buffalo, venison, kangaroo, squid, quail, rabbit, and salmon along with plant-based ingredients including tapioca, chickpeas, lentils, potato, and pumpkin. Total starch and gelatinized starch (as percent of total diet) were measured with variation due to ingredient type assessed using Student’s *t*-test in SAS 9.4. Significance was set at *p* < 0.05. Total starch (as a percent of diet) was higher in traditional diets compared to non-traditional diets formulated for maintenance (*p* < 0.0032) or all life stages (*p* < 0.0128). However, starch gelatinization as a proportion of total starch was lower in traditional diets formulated for maintenance (*p* < 0.0165) and all life stages (*p* < 0.0220). Total starch and gelatinized starch had a strong negative correlation (*r* = −0.78; *p* < 0.01) in diets utilizing traditional ingredients. These novel data reveal important differences between starch content and gelatinization and may impact selection of various ingredient types by pet food manufacturers.

## 1. Introduction

Non-traditional dietary formulations utilizing grain-free ingredient profiles for canines have become increasingly popular in recent years. These diets exclude traditional cereal grains but frequently contain other carbohydrates such as legumes or tubers. Legumes are frequently high in lysine, often a first-limiting amino acid, and as such are commonly utilized as complementary proteins to balance amino acid content [1,2]. Prior reports indicate lower dry matter (DM) and crude protein (CP) digestibility in the small intestine for diets formulated with increasing levels of soybean meal despite little impact to total tract digestibility in dogs [3]. Recent data have indicated that diets with grain-free formulations frequently contain legume content greater than 40% [2]. Data on finished, extruded pet foods are needed in order to assess the impact of manufacturing process on varying dietary formulation.

Unfortunately, digestibility studies are not plentiful in canine diets formulated with non-traditional ingredients. Although some data have reported digestibility potential for individual protein sources [4,5], more work is needed on non-traditional ingredients in finished products for canines to fully understand nutrient potential.

Prior characterization studies have examined total starch, resistant starch, and starch gelatinization in commercially available pet foods [6]. However, data for gelatinization in manufactured dog foods utilizing non-traditional diet formulations are lacking. Thus, the objective of this research was to measure starch content and gelatinization utilizing products formulated with both traditional and non-traditional ingredient profiles. Our hypothesis was that non-traditional dietary formulations with high inclusion rates of legumes and tubers will have differing gelatinization and total starch content when compared to diets formulated with traditional grains.

## 2. Materials and Methods

Twenty (*n* = 20) commercially available, dry extruded dog foods were purchased from two pet food suppliers (www.chewy.com and www.amazon.com) (accessed on 4 March 2019) during March 2019. Diets were selected based on ingredient profile and were classified as traditional (T), (*n* = 10) or non-traditional (NT), (*n* = 10) formulations. Traditional diets utilized animal-based ingredients including chicken, chicken by-product meal, meat and bone meal and plant-based ingredients including rice, barley, oats, and corn. Non-traditional formulations utilized meat-based ingredients including alligator, buffalo, venison, kangaroo, squid, quail, rabbit, and salmon along with plant-based ingredients including tapioca, chickpeas, lentils, potato, and pumpkin. Diets were further stratified within ingredient profile by life stage such that traditional diets included formulations for all life stages (*n* = 5) and for adult maintenance (*n* = 5). Similarly, non-traditional diets included formulations for all life stages (*n* = 5) and adult maintenance (*n* = 5). All diets selected were manufactured to meet or exceed NRC recommendations (see Table 1).

Each product was inspected upon receipt, whereupon sub-samples (50 g) were collected and shipped to a commercial analytical laboratory (Midwest Laboratories, Omaha, Nebraska; accreditation certificate number 2853.02). All diets were analyzed for three variables of interest. First, total starch (as a percentage of total diet) including all hydrolysable carbohydrates, was analyzed following a method adapted from AOAC 996.11 and YSI Application Note 319. Second, gelatinized starch (as a percentage of total diet) was measured using a method adapted from AOAC 996.11 [7]. Finally, total starch gelatinization (as % of total starch) was calculated using the following equation:$$\left\{\frac{\left(Gelatinized\;Starch\right)}{Total\;starch\;content}\right\}*\;100\%$$

Data were analyzed using SAS Studio (SAS Institute Inc., Cary, NC, USA). Student’s *t*-test was used to analyze total starch, starch gelatinization, and total gelatinized starch in non-traditional and traditional ingredient formulations for maintenance and all life stage diets. Linear regression and correlation were used to determine the relationship between total starch content and total gelatinized starch in traditional and non-traditional diets. Values are expressed as mean (±SD). Significance was set at *p* < 0.05.

## 3. Results

When ingredients were examined for the two diet formulation types, twenty-two animal and thirty-five plant sources were reported on product packaging (see Table 1). Non-traditional diets utilized plant-sourced ingredients as the first or second ingredient 30% (3/10) of the time while traditional formulations utilized plant ingredients in the first or second position 80% (8/10) of the time. Non-traditional diets utilized an animal ingredient as their first ingredient in all (10/10) products while traditional diets utilized animal ingredients first in only 80% (8/10) of the products examined.

Ingredient profile affected total starch as percent of diet in traditional (35.2 ± 4.0%) compared to non-traditional diets (22.3 ± 8.0%) formulated for adult maintenance (*p* = 0.0032) (Table 2).

Similarly, traditional ingredient profile also had increased total starch for diets manufactured for dogs of all life stages (36.0 ± 7.8%) compared to non-traditional formulations (19.9 ± 3.4%) (*p* = 0.0128).

When gelatinized starch (as % of total diet) was examined across both ingredient profiles, traditional diets were higher than non-traditional diets (30.2 ± 2.4% and 21.4 ± 7.8%, respectively) formulated for maintenance (*p* = 0.0165). Similarly, greater gelatinized starch was found in diets formulated for all life stages with traditional diets (30.9 ± 5.9%) compared to non-traditional diets (19.4 ± 3.4%) (*p* = 0.0220).

Proportionate amounts of total starch gelatinized was lower, however, for traditional diets than for non-traditional diets, although within those diets the values for total starch gelatinization were similar in both adult maintenance and all life stage formulations. Traditional ingredient formulations for maintenance were similar to all life stages (86.1 ± 3.8% and 86.3 ± 4.0%, respectively). Non-traditional diets had proportionately greater total starch gelatinization at 95.4 ± 2.1% and 97.6 ± 2.3% for maintenance and all life stages, respectively.

When the correlation between total starch content and gelatinized starch was examined, a strong negative relationship (*r* = −0.78; *p* < 0.01) was demonstrated in traditional diets, as shown in Figure 1A. However, no correlation (*r* = 0.28; *p* = 0.43) between total and gelatinized starch was measured in non-traditional diets (Figure 1B).

## 4. Discussion

The purpose of this work was to identify potential differences in starch gelatinization and total starch content in extruded dog foods based on ingredient profile. This study examined a limited number of extruded canine diets and found that non-traditional formulations had lower total starch and gelatinized starch, as percent of total diet, than traditional diets, but the percentage of starch gelatinized was greater in the non-traditional diets (~95% vs. ~85%). It is especially interesting to note that the negative correlation observed in traditional diets between gelatinization and total starch content was absent in non-traditional diets. Reasons for this difference are unclear but may include differences in amylose:amypectin ratio, degree of starch resistant to digestion, and other processing considerations.

In 2008, extruded dog foods were estimated to make up approximately 95% of dry pet foods [8]. The process of extrusion includes forcing raw ingredients under high pressure and temperature through a die for shaping and trimming. The process of extrusion results in the gelatinization of starches but may occur at varying temperatures based on other system factors (pressure, temperature, etc.) [6,9]. By altering temperature and moisture, the starch structure becomes disorganized and the granules begin to swell [10]. Starch gelatinization is reported to generate viscosity due to leaching of the amylose fraction, and allow expansion of kibble due to amylopectin content [10,11,12]. Because the crystalline structure of the amylose is disrupted during production, digestive enzymes have greater accessibility to the cellular structure of the starch granule, which may improve digestion [13]. The findings reported here may play a key role in improving our understanding of the potential energy available from diets formulated using non-traditional diets with high fractions of legume or tuber ingredients.

Although improved digestibility is commonly assumed with increased starch gelatinization, starch source can impact digestibility [14]. Prior work has demonstrated a direct relationship between percent gelatinization (also commonly referred to as % cook) and digestibility for some ingredients. Gelatinization improved digestibility of tapioca starch but had no effect on digestibility of wheat starch [15]. Peas and lentils had lower digestibility as compared to grains; however, these results were assessed from individual ingredients rather than the finished, final diet [16]. This may be due to the finely ground physical characteristics of the flours utilized in non-traditional diets as compared to traditional.

Other work comparing grain-free and ancient grain diets (including spelt, millet, sorghum) demonstrated full digestion of starch even with lower starch gelatinization for the ancient grain diet [17,18]. The biological significance of changes in digestibility may include significant impacts to blood glucose/insulin and colonic fermentation, which may impact VFA production. Highly digestible diets may increase glycemia and promote insulin release by the pancreas. Conversely, low digestibility may serve as a substrate for gastrointestinal microbiota with potential therapeutic impacts to insulin sensitivity [19].

The popularity of novel ingredients that is currently driving formulations including these ingredients must be approached systematically. Variation in components has significant impacts on gastrointestinal microbiota. Changes in microbial taxa have been reported for dogs offered raw meat diets with varying starch sources [20]. It is possible that lower digestibility coupled with higher levels of resistant starch impacted substrate availability in the large intestine, which likely impacted the resident microbiota. Future studies should evaluate differences related to starch gelatinization, resistant starch, fiber content, and fiber digestibility. 

The diets analyzed in our study differed in carbohydrate source. Grain-free diets have previously been found to have lower crude fiber, higher total dietary fiber (TDF), higher starch content, and a lower degree of processing [17]. These findings are dissimilar from the present study and may be due to the utilization of different grain sources and other manufacturing differences that cannot be controlled outside the laboratory. Many factors may affect total gelatinization, including amount of moisture, retention time, temperature, ingredient profile, and ingredient interactions [17,21]. Likely, the different ingredients (including carbohydrate type, coarseness of flour particulates, and degree of purification) along with other unexplored ingredient interactions contributed to the results presented here.

Future studies should include in vitro digestibility trials to determine the relationship between starch content, starch gelatinization, and digestibility for a wide variety of commonly utilized pet food ingredients, including those that are from non-traditional sources. Ultimately, comprehensive digestibility trials utilizing AAFCO approved standards should be completed in order to identify impacts to canine health associated with ingredient variation and interactions [22].

## 5. Conclusions

Although non-traditional dry extruded canine diets had less starch proportionally than traditional diets, the amount of starch gelatinized as a percentage of total starch was markedly higher in diets formulated with non-traditional ingredient profiles. Potential impacts of these differences are unknown and require future investigations. In conclusion, despite differences in total and gelatinized starch for traditional vs. non-traditional diets, no difference was measured based on life stage (adult maintenance vs. all life stage) despite markedly different nutrient requirements.

## Figures and Tables

**Figure 1 animals-12-03357-f001:**
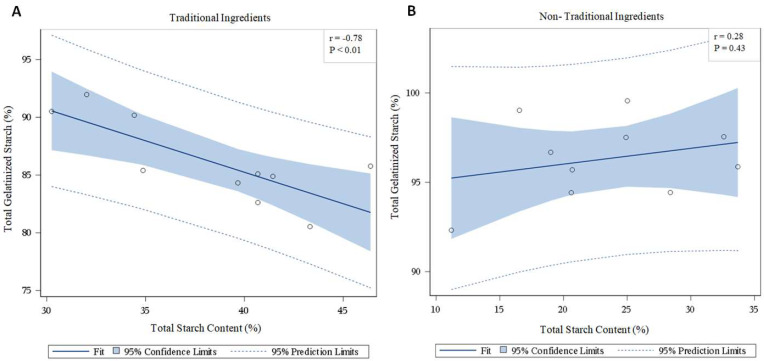
Correlation between starch content and total gelatinization in diets formulated with (**A**) traditional and (**B**) non-traditional ingredients.

**Table 1 animals-12-03357-t001:** Primary ingredients (listed in descending order of weight) in canine diets for maintenance and all life stages.

Diet	Type	Life Stage	Ingredient 1	Ingredient 2	Ingredient 3	Ingredient 4	Ingredient 5
1	NT	All life stages	Chicken	Chicken meal	Sweet potatoes	Pea protein	Pea flour
2	NT	All life stages	Deboned alligator	Menhaden fish meal	Tapioca starch	Peas	Pea protein
3	NT	All life stages	Kangaroo	Kangaroo meal	Peas	Chickpeas	Pea flour
4	NT	All life stages	Rabbit	Salmon meal	Menhaden fish meal	Chickpeas	Canola oil
5	NT	Maintenance	Buffalo	Lamb meal	Chicken meal	Sweet potatoes	Peas
6	NT	Maintenance	Chicken	Peas	Pea starch	Chicken by-product meal	Lentils
7	NT	Maintenance	Deboned venison	Turkey meal	Pork meal	Chickpeas	Lentils
8	NT	Maintenance	Quail	Chickpeas	Peas	Potatoes	Turkey meal
9	NT	Maintenance	Squid	Chickpeas	Pumpkin	Sunflower oil	Flaxseed
10	NT	Maintenance	Venison meal	Dried potatoes	Lentils	Chickpeas	Canola oil
11	T	All life stages	Chicken	Chicken meal	Whole grain brown rice	Cracked pearled barley	Pea flour
12	T	All life stages	Chicken meal	Brown rice	Rice	Chicken fat	Olive oil
13	T	All life stages	Chicken meal	Grain sorghum	Peas	Whole grain millet	Whole grain brown rice
14	T	Maintenance	Chicken	Brewers rice	Corn gluten meal	Whole grain corn	Poultry by-product
15	T	Maintenance	Chicken	Brown rice	Brewers rice	Cracked pearled barley	Chicken meal
16	T	Maintenance	Chicken	Chicken by-product meal	Corn meal	Ground whole grain sorghum	Brewers rice
17	T	Maintenance	Chicken	Organic barley	Organic oats	Organic peas	Chicken meal
18	T	Maintenance	Chicken meal	Ground barley	Ground oats	Ground brown rice	Chicken fat
19	T	Maintenance	Ground whole grain corn	Meat and bone meal	Corn gluten meal	Animal fat	Soybean meal
20	T	Maintenance	Whole ground brown rice	Dehydrated chicken	Coconut	Sun-cured alfalfa	Whole ground flaxseed

T = traditional ingredient profile. NT = non-traditional ingredient profile.

**Table 2 animals-12-03357-t002:** Starch content in non-traditional and traditional ingredient diets formulated for maintenance and all life stages.

	Non-Traditional (NT)	Traditional (T)	*p*-Value
** Maintenance **			
Total starch (% of total diet)	22.3 (±8.0)%	35.2 (±4.0)%	0.0032 *
Gelatinized starch (% of total diet)	21.4 (±7.8)%	30.2 (±2.4)%	0.0165 *
Total starch gelatinization (% of total starch)	95.4 (±2.1)%	86.1 (±3.8)%	0.0002 *
** All life stages **			
Total starch (% total diet)	19.9 (±3.4)%	36.0 (±7.8)%	0.0128 *
Gelatinized starch (% of total diet)	19.4 (±3.4)%	30.9 (±5.9)%	0.0220 *
Total starch gelatinization (% of total starch)	97.6 (±2.3)%	86.3 (±4.0)%	0.0049 *

Mean (±SD) * *p* < 0.05.

## Data Availability

The data provided in this study are available upon written request from the corresponding author.

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
