# Peer review of "An Assessment of Starch Content and Gelatinization in Traditional and Non-Traditional Dog Food Formulations"

_animals, 2022, doi:10.3390/ani12233357_

Round 1

Reviewer 1 Report (Previous Reviewer 1)

The manuscript has been revised and implemented in the missing parts.

Author Response

Thank you.

Reviewer 2 Report (Previous Reviewer 2)

The manuscript entitled "An Assessment of Starch Composition and Gelatinization in Traditional and Non-Traditional Dog Food Formulations" by Perry et al. tackles an interesting topic regarding starch content and its degree of gelatinization in traditional and non-traditional dog foods. The work has been revised in many parts and has acquired greater value for readers. It can be accepted without further revision.

Author Response

Thank you.

Reviewer 3 Report (New Reviewer)

This work has relevant information to the literature and had some interesting findings. However, these were not properly discussed and instead the authors discussed findings from other papers that mostly do not correlate to findings of the present study. Please see details below.

L45- substitute “potatoes” with “tubers”

L 47- balance

L 48- adverse impacts seem a very strong term. If the protein and DM digestibility were statistically lower, but not by more than 5%, I wouldn’t say it is impactful. It is just lower.

L 50-51 “Recent data has 50 indicated that current diets with grain-free formulations frequently contain legume con-51 tent greater than 40%” – please add a reference to that statement. 40% seems very high for legumes.

L 70-71- how did you make your purchasing decision? This section needs more detail.

L 91- missing SAS version, was it 9.4?

Table 2- Did you compare maintenance NT vs maintenance T, and all life stages NT vs all life stages T? If that was the case, your N was not 10. Please clarify under M&M. You could also include the sample size for each type of diet analyzed as a t-test (maintenance and all life stages)

Figure 1- very interesting finding!

L 139-140: “Extrusion is a quick process that utilizes high temperatures to produce kib-138 ble. This process results in the gelatinization of starches, denaturation of proteins, oxida-139 tion of lipids, and the Maillard Reaction”- This is poorly written. Extrusion is not just a high temperature process, there is also a lot of mechanical shear (mechanical energy) involved in the process. Please describe it with more detail. And you need a reference for the second phrase, because as far as I know oxidation of lipids and maillard reaction may happen, but are more prevalent in other steps of the process. For example, Maillard reaction is more common during drying, and lipid oxidation happens more commonly at the oil used for coating (in the presence of oxygen).

Paragraph L 150-159: it needs a conclusion- how does that relate to your work? I think this part needs to be better developed.

L 162- remove “some”. Gelatinization usually improves digestibility of all starches.

L 163-164: I recommend you revise biochemistry of starches. Amylopectin plays an important role in gelatinization, and yet you only mention amylose here. Amylose is the shorter linear glucose chain that leaches out upon gelatinization and increases viscosity. Amylopectin is branched and plays an important role in swelling of the starch granule. Please revise and add a reference.

L 165-167: This last statement is not very clear.

L170- a recent review of..?       

L173-175: Why do you write about TDF digestibility and then at the end you write “Those results are consistent with the findings from this study which found higher starch gelatinization in non-traditional diets.” How do those connect? Also, you discuss a lot about digestibility, although you haven’t measured any type of digestibility in your diets.

L180- It was a rough transition to start writing about microbial taxa all of a sudden, especially because it was not the focus of your work, and not measured. This paragraph also discussed another study, so how does that relate to your work?

Please discuss your findings. The discussion needs to be reformulated and has to have a logical sequence. You not once mentioned what is your hypothesis to why you had a high correlation between total starch and gelatinized starch just for traditional ingredients. You also ened to discuss more the findings in Table 2.

Round 2

Reviewer 3 Report (New Reviewer)

Please see my comments on the draft. The Discussion section must be improved in order to be considered for publication. Thank you.

Author Response

This manuscript is a resubmission of an earlier submission. The following is a list of the peer review reports and author responses from that submission.

Round 1

Reviewer 1 Report

In my opinion, the introduction should be revised and implemented for a better understanding of readers, it lacks concepts and should be enriched. Many concepts are instead present in the discussion which has more the appearance of an introduction.

The discussion seems very confusing and not very focused on the data obtained from this study.

The study is interesting but should be argued differently, so I suggest a major revision before publication.

Below are some comments:

Line 16: rabbit is repeated twice, it must be canceled.

Line 19: Has to be written Student’s t-test.

Lines 20-22: p-value must be written in italics and lowercase.

Lines 21-22: p-values you reported do not match those in the text, please revise them.

Lines 34-35: Please specify why the fact that these ingredients are high in lysine and low in methionine, allows their use as complementary proteins? Methionine is an essential amino acid for dogs as well as lysine.

Lines 38-39: Please explain how it happens that amino acids break down in that intestinal tract, and give references about this process.

Line 40: replace formulations with formulated.

Line 42: “are” instead of “is”.

Line 45: correct with “formulations”.

Line 49: please add “final” before products.

Line 60: Why only poultry? The references report several protein sources.

Lines 67-71: Highlight the importance of the purpose of this research in the final part of the introduction.

Line 68: Please specify the type of dog food, dry, wet, or semiwet.

Table 1: Specify whether the ingredients are listed in order of quantity in the final products.

Line 89: I would specify “Starch gelatinization (% on total starch) =” before the formula.

Line 90: is Student’s t-test.

Line 95: Brackets can be removed. Even if SD is known, enter the standard deviation in full. p-value must be written in italics and lowercase. Anyway, I think that the use of SEM is more appropriate for these data.

Line 98: is p.

Table 2: is p-value.

There are errors: “Starch gelatinization (% total starch) is not related to the % total starch but to the % of total diet, while the Total gelatinized starch percentage is related to the total starch content. Correct the table and check that all percentages calculated are rights.

Under the table I suggest to write “* p < 0.05” instead of “* = mean values differ P < 0.05

Lines 101-118: p is in italics and lowercase whenever it is mentioned.

Line 103: add the missing SD.

Line 107-112: Explain better what these data are and how you calculated them.

Figure 1: again is p

Lines 151-152: I think higher levels of dietary soluble fiber may result in a high level of digestibility as mentioned above, even if a better digestibility could be related to the gelatinization of course, but we are speaking of different things.

Author Response

Manuscript ID: Animals-1799244

Type:  Article

Title: An assessment of starch composition and gelatinization in traditional and non-traditional dog food formulations

Reviewer #1 - Response

In my opinion, the introduction should be revised and implemented for a better understanding of readers, it lacks concepts and should be enriched. Many concepts are instead present in the discussion which has more the appearance of an introduction.

Thank you for the opportunity to clarify.  We have reordered some of the material and have improved the clarity of these sections. 

The discussion seems very confusing and not very focused on the data obtained from this study.

The study is interesting but should be argued differently, so I suggest a major revision before publication.

Below are some comments:

Line 16: rabbit is repeated twice, it must be canceled.

revised

Line 19: Has to be written Student’s t-test.

revised

Lines 20-22: p-value must be written in italics and lowercase.

revised

Lines 21-22: p-values you reported do not match those in the text, please revise them.

Thank you for the opportunity to clarify.  It appears that the values reported on line 114 and in Figure 1A and 1B are both consistent with the p value for traditional (p<0.01) and non-traditional diets (p < 0.43).  Please identify the text that does not match the reported values. 

Lines 34-35: Please specify why the fact that these ingredients are high in lysine and low in methionine, allows their use as complementary proteins? Methionine is an essential amino acid for dogs as well as lysine.

Thank you for the opportunity to clarify.  We have revised the text for improved understanding by the reader (See line 34-35). 

Lines 38-39: Please explain how it happens that amino acids break down in that intestinal tract, and give references about this process.

We have revised the language to clarify.  Thank you for the opportunity improve the reader’s experience. 

Line 40: replace formulations with formulated.

Revised

Line 42: “are” instead of “is”.

Revised

Line 45: correct with “formulations”.

Revised

Line 49: please add “final” before products.

Revised

Line 60: Why only poultry? The references report several protein sources.

Revised

Lines 67-71: Highlight the importance of the purpose of this research in the final part of the introduction.

Revised

Line 68: Please specify the type of dog food, dry, wet, or semiwet.

Revised

Table 1: Specify whether the ingredients are listed in order of quantity in the final products.

Revised

Line 89: I would specify “Starch gelatinization (% on total starch) =” before the formula.

Revised

Line 90: is Student’s t-test.

Revised

Line 95: Brackets can be removed. Even if SD is known, enter the standard deviation in full. p-value must be written in italics and lowercase. Anyway, I think that the use of SEM is more appropriate for these data.

Thank you for the opportunity to address. The selection/reporting of SD in summary descriptive statistics was done to indicate the dispersion of data within the study group(s) as recommended by our statistical consultant.  We would agree that reporting of SEM is preferred to convey confidence in the reported mean value, but the mean value itself was not the focus for such heterogenous diets within the 2 compared groups (see Table 1).  Had the diets been more similar, the SEM might have been preferred.

Line 98: is p.

Revised

Table 2: is p-value.

Revised

There are errors: “Starch gelatinization (% total starch)” is not related to the % total starch but to the % of total diet, while the “Total gelatinized starch” percentage is related to the total starch content. Correct the table and check that all percentages calculated are rights.

Thank you for the opportunity to clarify.  The table is separated by diet type (maintenance and all life stages).  There are 3 rows within each diet type (total starch reported as a % of the total diet; starch gelatinization reported as a % of total starch; and total gelatinized starch which is a calculation of gelatinized starch divided by total starch.  Starch gelatinization is not reported as a % of the total diet here.  We have adjusted the descriptor following each row to clarify these data for the reader. 

Under the table I suggest to write “* p < 0.05” instead of “* = mean values differ P < 0.05”

Revised

Lines 101-118: is in italics and lowercase whenever it is mentioned.

Revised

Line 103: add the missing SD.

Revised

Line 107-112: Explain better what these data are and how you calculated them.

Figure 1: again is p

Revised

Lines 151-152: I think higher levels of dietary soluble fiber may result in a high level of digestibility as mentioned above, even if a better digestibility could be related to the gelatinization of course, but we are speaking of different things.

Thank you for the opportunity to clarify.  We have carefully examined the language of the manuscript and have made revisions to improve ease of understanding for the reader. 

Reviewer 2 Report

Reviewer: Comments and Suggestions for Authors

The manuscript entitled "An Assessment of Starch Composition and Gelatinization in Traditional and Non-Traditional Dog Food Formulations" by Perry et al. addresses an interesting topic regarding starch content and its degree of gelatinization in traditional and non-traditional dog foods. The work is well written and reasoned and represents a starting point for further studies such as the evaluation of the degree of digestibility in relation to the starch content.

Following are some comments which may be addressed before publication.

Decision: Minor Revision

Comment 1: I suggest replacing "starch content" in the Title and Table 2 rather than "starch composition" as in the manuscript the data reported concerns the content rather than the composition.

Comment 2: I suggest inserting the percentages of the various ingredients in Table 1 to better understand the different foods' composition.

Comment 3: Please correct the following typos:

Page 1, Line 16: Remove “rabbit”.

Page 1, Line 38: Add “a” before “little” and “the” before “total tract”.

Page 1, Line 42: change “is necessary” with “are necessary”.

Page 1, Line 45: correct “forumulations” with “formulations”

Page 5, Line 153: after “grain diets” remove “compared”. 

Author Response

Manuscript ID: Animals-1799244

Type:  Article

Title: An assessment of starch composition and gelatinization in traditional and non-traditional dog food formulations

Reviewer 2

The manuscript entitled "An Assessment of Starch Composition and Gelatinization in Traditional and Non-Traditional Dog Food Formulations" by Perry et al. addresses an interesting topic regarding starch content and its degree of gelatinization in traditional and non-traditional dog foods. The work is well written and reasoned and represents a starting point for further studies such as the evaluation of the degree of digestibility in relation to the starch content.

Following are some comments which may be addressed before publication.

Decision: Minor Revision

 Comment 1: I suggest replacing "starch content" in the Title and Table 2 rather than "starch composition" as in the manuscript the data reported concerns the content rather than the composition.

Revised as suggested.

Comment 2: I suggest inserting the percentages of the various ingredients in Table 1 to better understand the different foods' composition.

Thank you for the opportunity to clarify.  Unfortunately, without access to each of the manufacturer’s proprietary formulation data, we have no way to report individual inclusion rates for each of the ingredients.  We have revised the text to include a description that the ingredients are listed in descending order. 

Comment 3: Please correct the following typos:

Page 1, Line 16: Remove “rabbit”.

Revised

Page 1, Line 38: Add “a” before “little” and “the” before “total tract”.

Text revised

Page 1, Line 42: change “is necessary” with “are necessary”.

Text revised

Page 1, Line 45: correct “forumulations” with “formulations”

Revised

Page 5, Line 153: after “grain diets” remove “compared”. 

Revised

Reviewer 3 Report

Dear authors,

the manuscript investigates an interesting research question and it is nice that you are sharing preliminary results. However, the sample size and performed analysis are not sufficient for an original research article.  You may want to perform a more advanced study or publish the results as short communication instead.

Here are some additional remarks:

-check the entire manuscript for double spaces, correct spelling and grammatical errors (e.g. line 10,  

-add information on how the microbiota (not microflora, that would be plants) is affected when the ileal digestibility is reduced

-Line 60: why poultry?

-make sure not to mix up the terms grain and carbohydrate source

-no need to mention where the feed was bought.

-Are the diets all complete diets?

-define traditional/non-traditional better. There are some samples that contain peas, but ar considered traditional (13) while others are considered non traditional (6)

-table 1 needs to be formatted

-were there differences in DM? is Table 2 referring to DM or as fed? What is the unit for total gelatinized starch?

-what do you mean by ancient grain?

-Why do soluble fibres result in higher gelatinization?

-how is the digestibility linked to gelatinization? (line 134)

Line 162: it should also investigate if protein digestibility is affected

Sort references

Author Response

Manuscript ID: Animals-1799244

Type:  Article

Title: An assessment of starch composition and gelatinization in traditional and non-traditional dog food formulations

Reviewer #3

Dear authors,

the manuscript investigates an interesting research question and it is nice that you are sharing preliminary results. However, the sample size and performed analysis are not sufficient for an original research article.  You may want to perform a more advanced study or publish the results as short communication instead.

Here are some additional remarks:

-check the entire manuscript for double spaces, correct spelling and grammatical errors (e.g. line 10,  

Thank you for the opportunity to revise.  We have carefully examined the manuscript and have edited the text for improved ease of reading. 

-add information on how the microbiota (not microflora, that would be plants) is affected when the ileal digestibility is reduced

Revised

-Line 60: why poultry?

Revised to remove the reference to poultry

-make sure not to mix up the terms grain and carbohydrate source

Thank you for the opportunity to clarify.  Although there were different grain sources utilized, the use of tubers and legumes may have also contributed to starch content and thus impact variables of interest.  We have carefully examined the language and have taken care to specifically utilize "grain" or "carbohydrate" purposefully when referencing these ingredient types.  

-no need to mention where the feed was bought.

Revised

-Are the diets all complete diets?

All diets were formulated for either “Maintenance” or “All Life Stages”. 

-define traditional/non-traditional better. There are some samples that contain peas, but ar considered traditional (13) while others are considered non traditional (6)

Thank you for the opportunity to clarify.  We have edited the text to remove “peas” for improved separation between the two dietary ingredient lists (See line 75-80).

-table 1 needs to be formatted

Thank you.  We’ve been instructed to insert the tables as they are mentioned within the text.  The publishing editors will adjust the visual appearance of the tables as appropriate. 

-were there differences in DM? is Table 2 referring to DM or as fed? What is the unit for total gelatinized starch?

Thank you for the opportunity to clarify.  We have adjusted the nomenclature with an expanded definition(See lines 85-90) and have adjusted the rows within the table to better identify each of the variables of interest. 

-what do you mean by ancient grain?

Thank you for the opportunity to clarify.  We have added text to identify the grains in question. 

-Why do soluble fibres result in higher gelatinization?

Revised for clarity.

-how is the digestibility linked to gelatinization? (line 134)

Thank you for the opportunity to clarify.  We have adjusted the wording of the Discussion and changed some of the content in the Introduction to better address this question. 

Line 162: it should also investigate if protein digestibility is affected

Agree!  We have studies underway to understand impacts to amino acid availability.

Sort references

Revised.

Round 2

Reviewer 1 Report

The paper has been barely improved, but not enough to be considered acceptable, many points remain to be clarified, many parts have been removed but have not been implemented and many topics have not been addressed.

Furthermore, the conclusions do not have the appearance of real conclusions and should all be rewritten. Therefore I still suggest major revisions before accepting this paper.

Below are some comments:

Line 32: Again, is Student’s t-test

Lines 34-35: Again, the p values p <0.0001 do not match those in the text and table 2.

Line 49: Why lysine is so important compared to other amino acids, please specify.

Table 2:

A: Thank you for the opportunity to clarify. The table is separated by diet type (maintenance and all life stages). There are 3 rows within each diet type (total starch reported as a % of the total diet; starch gelatinization reported as a % of total starch; and total gelatinized starch which is a calculation of gelatinized starch divided by total starch. Starch gelatinization is not reported as a % of the total diet here. We have adjusted the descriptor following each row to clarify these data for the reader.

R: What you asserted is not correct as you report a percentage, if it were only the ratio of gelatinized starch divided by total starch the values would be 0.9596 and 0.9749, while you expressed it as a percentage as you divided the gelatinized starch by total starch and multiply it by 100, please revise all.

Lines 152-159: Again, explain how you calculate these values (these are %) for a better understanding of readers.

Reviewer 3 Report

Dear authors,

thank you for the revised manuscript. Unfortunately, many comments were not addressed and while the language has been improved, little has been done to improve its content. As example, the entire section " However, the sample size and performed analysis are not sufficient for an original research article.  You may want to perform a more advanced study or publish the results as short communication instead." did not get any response. You could e.g. add some in vitro digestibility results to improve the manuscript. Also, Table 2 is still missing the unit for starch gelatinization, if % refers to DM or as fed, and it is still not clear if the diets were complete diets or not (yes, it is stated if they are mend for all life stages or for maintenance, but not if complete or not). This could also be investigated (are the diets balanced?) to improve the manuscript. The term ancient grains was still not explained, even though you state that you did and the association between soluble DF and digestibility was not clarified. I suggest to wait for your additional results before publishing.

Additional comments:

Line 206-208. This does not make sence. First you state that TDFs are not digested but later you write that these end products from fermentation (which ones??) increase TDF digestibility. Do you mean TDF disaperance or fermentation?

The conclusions should conclude your findings, not summarize what has been done and still has to be done (by other studies).